# Training Large Language Models to Reason in a Continuous Latent Space

## Abstract

Large language models are restricted to reason in the "language space", where they typically express the reasoning process with a chain-of-thoughts (CoT) to solve a complex reasoning problem. However, we argue that language space may not be the optimal reasoning space. For example, most word tokens are primarily for textual coherence and not essential for reasoning, while some critical tokens require complex planning and pose huge challenges to LLMs. To explore the potential of LLM reasoning in an unrestricted latent space instead of using human language, we introduce a new paradigm COCONUT (**C**hain **o**f **Con**tin**u**ous **T**hought). We utilize the last hidden state of the LLM as a representation of the reasoning state (termed "continuous thought"). Rather than decoding this into a word token, we feed it back to the LLM as the subsequent input embedding directly in the continuous space. Experiments show that COCONUT can effectively augment the LLM on several reasoning tasks. It even outperforms CoT in certain logical reasoning tasks that require substantial planning, despite generating fewer tokens during inference. More interestingly, we observe an advanced reasoning patterns emerging from latent reasoning: the continuous thought can encode multiple potential next reasoning steps, allowing the model to perform a breadth-first search (BFS) to solve the problem, rather than prematurely committing to a single deterministic path like CoT. These findings demonstrate the promise of latent reasoning and offer valuable insights for future research on latent reasoning methods.

## 1 Introduction

Large language models (LLMs) have demonstrated remarkable reasoning abilities, emerging from extensive pretraining on human language (Dubey et al., 2024; Achiam et al., 2023). While the next token prediction is an effective training objective, it imposes a fundamental constraint pn the LLM as a reasoning machine: the reasoning process of LLMs must be generated in word tokens. For example, a prevalent approach, known as chain-of-thought (CoT) reasoning (Wei et al., 2022), involves prompting or training LLMs to generate solutions step-by-step using natural language. However, this stands in stark contrast to human cognition. Neuroimaging studies have consistently shown that the language network – a set of brain regions responsible for language comprehension and production – remains largely inactive during various reasoning tasks (Amalric & Dehaene, 2019; Monti et al., 2012; 2007; 2009; Fedorenko et al., 2011). More evidence has indicated that human language is optimized for communication rather than reasoning (Fedorenko et al., 2024).

A significant problem arises when LLMs are required to output language during reasoning: the "reasoning amount" behind each token varies greatly, yet current LLM architectures allocate nearly the same computing budget for predicting every token. Most tokens in a reasoning chain are generated solely for fluency, contributing little to the actual reasoning process. On the contrary, some critical tokens require complex planning and pose huge challenges to LLMs. While previous work has attempted to fix these problems by prompting LLMs to generate succinct reasoning chains (Madaan & Yazdanbakhsh, 2022), or performing additional reasoning before generating some critical tokens (Zelikman et al., 2024), these solutions remain constrained within the language space and do not solve the problems fundamentally. Ideally, LLMs should be allowed to reason freely in an unconstrained latent space and only translate the outcomes into language once the reasoning process is complete.

Figure 1: A comparison of CoT and COCONUT. In CoT, the model generates the reasoning process as a word token sequence (e.g., $[x_i, x_{i+1}, ..., x_{i+j}]$ in the figure). COCONUT (Chain of Continuous Thoughts) regards the last hidden state as a representation of reasoning state (termed "continuous thought"), and directly uses it as the next input embedding. This allows the LLM to reason in an unrestricted latent space instead of language space.

We aim to explore LLM reasoning in the latent space by introducing a novel paradigm, COCONUT (Chain of Continuous Thought). It involves a simple modification to the traditional CoT process. Instead of mapping between hidden states and language tokens using the language model head and embedding layer, COCONUT directly feeds the last hidden state (a continuous thought) as the input embedding for the next token (Figure 1). This modification frees the reasoning from language space, and the architecture can be optimized end-to-end by gradient descent, as continuous thoughts are fully differentiable. To enhance the training of these continuous thoughts, we employ a multi-stage training strategy inspired by Deng et al. (2024), which effectively utilizes language reasoning chains to guide the training process.

The experiments demonstrate that COCONUT successfully enhances the reasoning capabilities of LLMs. Specifically, on math reasoning problems (GSM8k, Cobbe et al., 2021), using more continuous thoughts is shown to be beneficial to reasoning accuracy, mirroring the effects of language reasoning chains. This indicates the potential to scale and solve increasingly challenging problems by chaining more continuous thoughts. On logical reasoning problems including ProntoQA (Saparov & He, 2022), and our newly proposed ProsQA (Section 4.1) which requires stronger planning ability, COCONUT and some of its variants even surpasses language-based CoT methods, while generating significantly fewer tokens during inference.

Interestingly, the removal of language space constraints has led to a novel reasoning pattern. By manipulating the COCONUT model to switch between latent reasoning and language reasoning, we are able to unveil the latent reasoning process. Unlike language-based reasoning, continuous thoughts in COCONUT can encode multiple potential next steps simultaneously, allowing for a reasoning process akin to breadth-first search (BFS). While the model may not initially make the correct decision, it can maintain all possible options within the continuous thoughts and progressively eliminate incorrect paths through reasoning, guided by some implicit value functions. This advanced reasoning mechanism surpasses traditional CoT approaches, even though the model is not explicitly trained or instructed to operate in this manner, as seen in previous works (Yao et al., 2023; Hao et al., 2023). We believe that these findings underscore the potential of latent reasoning and could provide valuable insights for future research.

## 2 RELATED WORK

**Chain-of-thought (CoT) reasoning.** We use the term chain-of-thought broadly to refer to methods that generate an intermediate reasoning process in language before outputting the final answer. This includes prompting LLMs (Wei et al., 2022; Khot et al., 2022; Zhou et al., 2022), or training LLMs to generate reasoning chains, either with supervised fine-tuning (Yue et al., 2023; Yu et al., 2023) or reinforcement learning (Wang et al., 2024; Havrilla et al., 2024; Shao et al., 2024; Yu et al., 2024a). Madaan & Yazdanbakhsh (2022) classified the tokens in CoT into symbols, patterns, and text, and proposed to guide the LLM to generate concise CoT based on analysis of their roles. Recent theoretical analyses have demonstrated the usefulness of CoT from the perspective of model expressivity (Feng et al., 2023; Merrill & Sabharwal, 2023; Li et al., 2024). By employing CoT, the effective depth of the transformer increases because the generated outputs are looped back to the input (Feng et al., 2023). These analyses, combined with the established effectiveness of CoT, mo-

tivated our exploration of continuous thoughts, in contrast to other latent reasoning methods. While CoT has proven effective for certain tasks, its autoregressive generation nature makes it challenging to mimic human reasoning on more complex problems (LeCun, 2022; Hao et al., 2023), which typically require planning and search. There are works that equip LLMs with explicit tree search algorithms (Xie et al., 2023; Yao et al., 2023; Hao et al., 2023), or train the LLM on search dynamics and trajectories (Lehnert et al., 2024; Gandhi et al., 2024). In our analysis, we find that after removing the constraint of language space, a new reasoning pattern similar to BFS emerges, even though the model is not explicitly trained in this way.

**Latent reasoning of LLM.** Previous works mostly define latent reasoning of LLM as the hidden computing in transformers (Yang et al., 2024; Biran et al., 2024). Yang et al. (2024) constructed a dataset of two-hop reasoning problems and discovered that it is possible to recover the intermediate variable from the hidden representation of LLMs. Biran et al. (2024) further proposed to intervene the latent reasoning by "back-patching" the hidden representation. Another line of work has discovered that, even if the model generates a CoT to reason, the model may actually utilize a different latent reasoning process. This phenomenon is known as the unfaithfulness of CoT reasoning (Wang et al., 2022; Turpin et al., 2024). To enhance the latent reasoning of LLM, previous research proposed to augment it with additional tokens. Goyal et al. (2023) pretrained model by randomly inserting a learnable `<pause>` tokens to the corpus. This improves LLM's performance on a variety of tasks, especially when followed by supervised finetuning with `<pause>` tokens. On the other hand, Pfau et al. (2024) further explored the usage of filler tokens, e.g., "...", and concluded that they work well for highly parallelizable problems. However, these methods do not extend the expressivity of the LLM like CoT (Pfau et al., 2024); hence, they may not scale to more general and complex reasoning problems. Recently, it has also been found that one can "internalize" the chain of thought reasoning into latent reasoning with knowledge distillation (Deng et al., 2023) or a special training curriculum which gradually shortens CoT (Deng et al., 2024). Yu et al. (2024b) also proposed to distill a model that can reason latently from data generated with complex reasoning algorithms. These training methods can be combined to our framework, and specifically, we find that breaking down the learning of continuous thoughts into multiple stages, inspired by iCoT (Deng et al., 2024), is very beneficial for the training.

## 3 COCONUT: CHAIN OF CONTINUOUS THOUGHTS

In this section, we introduce our new paradigm COCONUT (Chain of Continuous Thoughts) for reasoning outside the language space. We begin by introducing the background and notations of language models. For an input sequence $x = (x_1, ..., x_T)$, the standard large language model $\mathcal{M}$ can be described as:

$$H_t = \text{Transformer}(E_t + P_t)$$
$$\mathcal{M}(x_{t+1} \mid x_{\leq t}) = \text{softmax}(W h_t)$$

where $E_t = [e(x_1), e(x_2), ..., e(x_t)]$ is the sequence of token embeddings up to position $t$; $P_t = [p(1), p(2), ..., p(t)]$ is the sequence of positional embeddings up to position $t$; $H_t \in \mathbb{R}^{t \times d}$ is the matrix of the last hidden states for all tokens up to position $t$; $h_t$ is the last hidden state of position $t$, i.e., $h_t = H_t[t, :]$; $e(\cdot)$ is the token embedding function; $p(\cdot)$ is the positional embedding function; $W$ is the parameter of the language model head.

**Method Overview.** In the proposed COCONUT method, the LLM switches between the "language mode" and "latent mode" (Figure 1). In language mode, the model operates as a standard language model, autoregressively generating the next token. In latent mode, it directly utilizes the last hidden state as the next input embedding. This last hidden state represents the current reasoning state, termed as a "continuous thought".

Special tokens `<bot>` and `<eot>` are employed to mark the beginning and end of the latent mode, respectively. As an example, we assume latent reasoning occurs between positions $i$ and $j$, i.e., $x_i = $ `<bot>` and $x_j = $ `<eot>`. When the model is in the latent mode $(i < t < j)$, we use the last hidden state from the previous token to replace the input embedding, i.e., $E_t = [e(x_1), e(x_2), ..., e(x_i), h_i, h_{i+1}, ..., h_{t-1}]$. After the latent mode finishes, $(t \geq j)$, the input after position reverts to using the token embedding, i.e., $E_t = $

Figure 2: The training procedure of COCONUT. At each stage, we integrate $c$ additional continuous thought ($c = 1$ in this example), and remove one reasoning step in the training data. The cross-entropy loss is then calculated on the remaining tokens after continuous thoughts. $[e(x_1), e(x_2), ..., e(x_i), h_i, h_{i+1}, ..., h_{j-1}, e(x_j), ..., e(x_t)]$. It is noteworthy that $\mathcal{M}(x_{t+1} \mid x_{\leq t})$ is not defined when $i < t < j$, since the latent thought is not intended to be mapped back to language space. However, $\text{softmax}(Wh_t)$ can still be calculated for probing purposes (see Section 4).

**Training Procedure.** In this work, we focus on a problem-solving setting where the model receives a question as input and is expected to generate an answer through a reasoning process. We leverage language CoT data to supervise continuous thought by implementing a multi-stage training curriculum inspired by Deng et al. (2024). As shown in Figure 2, in the initial stage, the model is trained on regular CoT instances. In the subsequent stages, at the $k$-th stage, the first $k$ reasoning steps in the CoT are replaced with $k \times c$ continuous thoughts[1], where $c$ is a hyperparameter controlling the number of latent thoughts replacing a single language reasoning step. Following Deng et al. (2024), we also reset the optimizer state when training stages switch. We insert <bot> and <eot> tokens to encapsulate the continuous thoughts.

During the training process, we mask the loss on questions and latent thoughts. It is important to note that the objective does not encourage the continuous thought to *compress the removed language thought*, but rather to *facilitate the prediction of future reasoning*. Therefore, it's possible for the LLM to learn a more effective representation compared to language reasoning steps.

**Training Details.** Our proposed continuous thoughts are fully differentiable, allowing backpropagation. We perform $n + 1$ forward passes when $n$ latent thoughts are scheduled in the current training stage, computing a new latent thought with each pass and then conducting an additional forward pass to obtain a loss on the remaining text sequence. While we can save any repetitive computing by using KV cache, the sequential nature of the multiple forward passes poses challenges for parallelism. Further optimizing the training efficiency of COCONUT remains an important direction for future research.

**Inference Process.** The inference process for COCONUT is analogous to standard language model decoding, except that in latent mode, we directly feed the last hidden state as the next input embedding. A challenge lies in determining when to switch between latent and language modes. As we focus on the problem-solving setting, we insert a <bot> token immediately following the question tokens. For <eot>, we consider two potential strategies: a) train a binary classifier on latent thoughts to enable the model to autonomously decide when to terminate the latent thoughts, or b) always pad the latent thoughts to a constant length. We found that both approaches work comparably well. Therefore, we use the second option in our experiment for simplicity, unless specified otherwise.

## 4 EXPERIMENTS

In this section, we validate the feasibility of LLM reasoning in latent space through experiments on three datasets. We mainly evaluate the accuracy by comparing the model-generated answers with the ground truth. The number of newly generated tokens per question is also listed, as a measure of reasoning efficiency.[2]

---

[1]If a reasoning chain is shorter than $k$ steps, then all the language thoughts will be removed.

[2]One continuous thought is counted as one token since the computational cost is essentially the same.

### 4.1 DATASETS

**Math Reasoning.** We use GSM8k (Cobbe et al., 2021) as the dataset for math reasoning. It consists of grade school-level math problems. Compared to other datasets of our experiments, the problems are more diverse and open-domain, closely resembling the real-world use cases. Through this task, we explore the potential of latent reasoning in practical applications. To train the model, we use a synthetic dataset generated by Deng et al. (2023).

**Logical Reasoning.** Logical reasoning involves the proper application of known conditions to prove or disprove a conclusion using logical rules. This requires the model to choose from multiple possible reasoning paths, where the correct decision often relies on exploration and planning ahead. This serves as a simplified simulation of more advanced reasoning tasks, such as automated theorem proving (Chen et al., 2023; DeepMind, 2024). We use 5-hop ProntoQA (Saparov & He, 2022) questions, with fictional concept names. For each problem, an tree-structured ontology is randomly generated and described in natural language as a set of known conditions. The model is asked to judge whether a given statement is correct based on these conditions.

We found that the generation process of ProntoQA was overly simplified, especially since the size of distracting branches in the ontology is always small, reducing the need for complex planning. To fix that, we apply a new dataset construction pipeline using randomly generated DAGs to structure the known conditions. The resulting dataset requires the model to perform substantial planning and searching over the graph to find the correct reasoning chain. We refer to this new dataset as the ProsQA (**Pro**of with **S**earch **Q**uestion-**A**nswering). A visualized example is shown in Figure 6. More details of datasets can be found in Appendix A.

### 4.2 EXPERIMENTAL SETUP

We pre-trained GPT-2 (Radford et al., 2019) as the base model for all experiments. The learning rate is set to $1 \times 10^{-4}$ while the effective batch size is 128. Following Deng et al. (2024), we also reset the optimizer when training stages switch.

**Math Reasoning.** By default, we use 2 latent thoughts (i.e., $c = 2$) for each reasoning step. we analyze the correlation between performance and $c$ in Section 4.4. The model goes through 3 stages besides the initial stage. Then, we will have an additional stage, where we still use $3 \times c$ continuous thoughts as in the last stage, but remove all the remaining language reasoning chain. This handles the long-tail distribution of reasoning chains longer than 3 steps. We train the model for 6 epochs in the initial stage, and 3 epochs in each remaining stage.

**Logical Reasoning.** We use one continuous thought for every reasoning step (i.e., $c = 1$). The model goes through 6 training stages in addition to the initial stage, because the maximum number of reasoning steps is 6 in these two datasets, and the model fully reasons with continuous thoughts to solve the problems in the last stage. We train the model for 5 epochs per stage.

For all datasets, after the standard schedule, the model stays in the final training stage, until the 50th epoch. We select the checkpoint based on the accuracy on the validation set. For inference, we manually set the number of continuous thoughts to be consistent with their final training stage. We use greedy decoding for all experiments.

### 4.3 BASELINES AND ABLATIONS

We consider the following baselines: (1) *CoT*: We use the complete reasoning chains to train the language model with supervised finetuning, and during inference, the model generates a reasoning chain before outputting an answer. (2) *No-CoT*: The LLM is trained to directly generate the answer without using a reasoning chain. (3) *iCoT* (Deng et al., 2024): The model is trained with language reasoning chains and follows a carefully designed schedule that "internalizes" CoT. As the training goes on, tokens at the beginning of the reasoning chain are gradually removed until only the answer remains. During inference, the model directly predicts the answer. (4) *Pause token* (Goyal et al., 2023): The model is trained using only the question and answer, without a reasoning chain. However, different from *No-CoT*, special <pause> tokens are inserted between the question and answer, which are believed to provide the model with additional computational capacity to derive

| Method | GSM8k | | ProntoQA | | ProsQA | |
|---|---|---|---|---|---|---|
| | Acc. (%) | # Tokens | Acc. (%) | # Tokens | Acc. (%) | # Tokens |
| CoT | 42.9 ±0.2 | 25.0 | 98.8 ±0.8 | 92.5 | 77.5 ±1.9 | 49.4 |
| No-CoT | 16.5 ±0.5 | 2.2 | 93.8 ±0.7 | 3.0 | 76.7 ±1.0 | 8.2 |
| iCoT | 30.0* | 2.2 | 99.8 ±0.3 | 3.0 | 98.2 ±0.3 | 8.2 |
| Pause Token | 16.4 ±1.8 | 2.2 | 77.7 ±21.0 | 3.0 | 75.9 ±0.7 | 8.2 |
| COCONUT (Ours) | 34.1 ±1.5 | 8.2 | 99.8 ±0.2 | 9.0 | 97.0 ±0.3 | 14.2 |
| - w/o curriculum | 14.4 ±0.8 | 8.2 | 52.4 ±0.4 | 9.0 | 76.1 ±0.2 | 14.2 |
| - w/o thought | 21.6 ±0.5 | 2.3 | 99.9 ±0.1 | 3.0 | 95.5 ±1.1 | 8.2 |
| - pause as thought | 24.1 ±0.7 | 2.2 | 100.0 ±0.1 | 3.0 | 96.6 ±0.8 | 8.2 |

Table 1: Results on three datasets. Higher accuracy indicates stronger reasoning ability, while generating fewer tokens indicates better efficiency. *The result of *iCoT* is from Deng et al. (2024).

the answer. For a fair comparison, the number of `<pause>` tokens is set the same as continuous thoughts in COCONUT.

We also evaluate some variants of our method: (1) *w/o curriculum*: Instead of the multi-stage training, we directly use the data from the last stage which only includes questions and answers to train COCONUT. The model uses continuous thoughts to solve the whole problem. (2) *w/o thought*: We keep the multi-stage training which removes initial reasoning steps gradually, but don't use any continuous latent thought. While this is similar to *iCoT* in the high-level idea, the exact training schedule is set to be consistent with COCONUT, instead of *iCoT*. This ensures a more strict comparison. (3) *Pause as thought*: We use special `<pause>` tokens to replace the continuous thought, and apply the same multi-stage training scheme as COCONUT.

## 4.4 RESULTS AND DISCUSSION

We show the overall results on all datasets in Table 1. Continuous thoughts effectively enhance LLM reasoning, as shown by the consistent improvement over *no-CoT*. It even shows better performance than *CoT* on ProsQA. We describe several key conclusions from the experiments as follows.

**"Chaining" continuous thoughts enhances reasoning.** In conventional CoT, the output token serves as the next input, which is believed to increase the effective depth of LLMs and enhance their expressiveness (Feng et al., 2023). We explore whether latent space reasoning retains this property, as it would suggest that this method could scale to solve increasingly complex problems by chaining multiple latent thoughts.

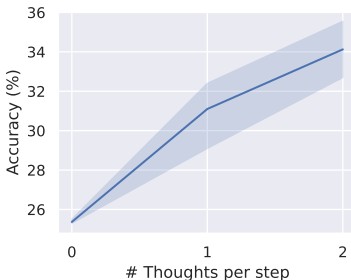

Figure 3: Accuracy on GSM8k with different number of continuous thoughts.

In our experiments with GSM8k, we found that COCONUT outperformed other architectures trained with similar strategies, particularly surpassing the latest baseline, *iCoT* (Deng et al., 2024). The performance is significantly better than CO-CONUT (*pause as thought*) which also enables more computation in the LLMs. While Pfau et al. (2024) empirically shows that filler tokens, such as the special `<pause>` tokens, can benefit highly parallelizable problems, our results show that COCONUT architecture is more effective for general problems, e.g., math word problems, where a reasoning step often heavily depends on previous steps. Additionally, we experimented with adjusting the hyperparameter $c$, which controls the number of latent thoughts corresponding to one language reasoning step. As we increased $c$ from 0 to 1 to 2, the model's performance steadily improved (Figure 3). These results strongly suggest that a chaining effect similar to CoT can be observed in the latent space.

In two other synthetic tasks, we found that the varients of COCONUT (*w/o thoughts* or *pause as thought*), and *iCoT* also achieve impressive accuracy. This indicates that in these tasks, the model's computational capacity may not the bottleneck. In contrast, GSM8k, being an open-domain question-answering task, likely involves more complex contextual understanding and modeling, placing higher demands on computational capability.

**Latent Reasoning Excels Language Reasoning in Planning.** Some complex reasoning tasks require the model to "look ahead" to assess whether a particular step is the right choice. Among the

datasets used in our experiments, GSM8k consists of grade-school-level math word problems, allowing for intuitive judgment of the next reasoning step; ProntoQA has distracting branches of small sizes, which makes it relatively easy to determine the next step too. In contrast, ProsQA is based on a randomly generated DAG structure, posing a significant challenge to the model's planning abilities. Reasoning in language space cannot effectively solve the problem. As shown in the table, *CoT* doesn't show significant improvement over *No-CoT*. On the contrary, COCONUT, some of its variants and *iCoT* significantly improve the reasoning on ProsQA. This suggests an advantage in using latent space over language tokens for tasks requiring extensive planning. We conduct in-depth analysis of the latent reasoning process in Section 5.

**The LLM still needs guidance to learn continuous thoughts.** In the ideal case, the model should learn the most effective continuous thoughts automatically through gradient descent on questions and answers (i.e., COCONUT *w/o curriculum*). However, from the experimental results, we found the models trained this way do not perform any better than no-CoT.

With the multi-stage curriculum which decomposes the training into easier objectives, CO-CONUT is able to achieve top performance across various tasks. The multi-stage training also integrates well with pause tokens (COCONUT- *pause as thought*). Despite using the same architecture and similar multi-stage training objectives, we observed a small gap between the performance of *iCoT* and CO-CONUT (*w/o thoughts*). The finer-grained removal schedule (token by token) and a few other tricks in *iCoT* may ease the training process. We leave combining *iCoT* and COCONUT as a future work. While the multi-stage training used for COCONUT has proven effective, further research is definitely needed to develop better and more general strategies for learning reasoning in latent space, especially without the supervision from language reasoning chains.

Figure 4: A case study where we decode the continuous thought into language tokens

**Continuous thoughts are efficient representations of reasoning.** Though the continuous thoughts are not intended to be decoded to language tokens, we can still use it as an intuitive interpretation of the latent reasoning. We show a case study in Figure 4 of a math word problem solved by COCONUT ($c = 1$). The first continuous thought can be decoded into tokens like "180", " 180" (with a space), and "9". Note that, the reasoning trace for this problem should be $3 \times 3 \times 60 = 9 \times 60 = 540$, or $3 \times 3 \times 60 = 3 \times 180 = 540$. The interpretations of the first thought happen to be the first intermediate variables in the calculation. Moreover, it encodes a distribution of different traces into the continuous thoughts. As shown in Section 5.3, this feature enables a more advanced reasoning pattern for planning-intense reasoning tasks.

## 5 UNDERSTANDING THE LATENT REASONING IN COCONUT

In this section, we present an analysis of the latent reasoning process with a variant of COCONUT. By leveraging its ability to switch between language and continuous space reasoning, we are able to control the model to interpolate between fully latent reasoning and fully language reasoning and test their performance (Section 5.2). This also enables us to interpret the the latent reasoning process as tree search (Section 5.3). Based on this perspective, we explain why latent reasoning can make the decision easier for LLMs (Section 5.4).

### 5.1 EXPERIMENTAL SETUP

**Methods.** The design of COCONUT allows us to control the number of latent thoughts by manually setting the position of <eot> token during inference. When we enforce COCONUT to use $k$

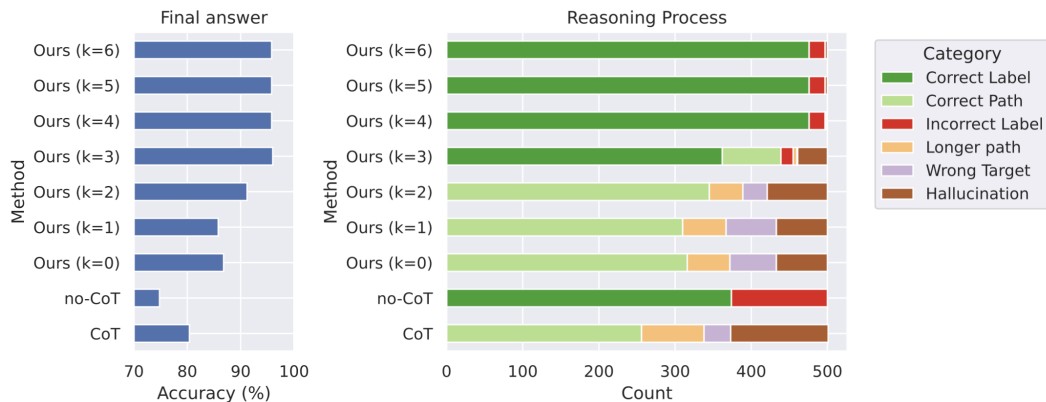

Figure 5: The accuracy of final answer (left) and reasoning process (right) of multiple varients of COCONUT and baselines on ProsQA.

continuous thoughts, the model is expected to output the remaining reasoning chain in language, starting from the $k + 1$ step. In our experiments, we test variants of COCONUT on ProsQA with $k \in \{0, 1, 2, 3, 4, 5, 6\}$. Note that all these variants only differ in inference time while sharing the same model weights. Besides, we report the performance of *CoT* and *no-CoT* as references.

To address the issue of forgetting earlier training stages, we modify the original multi-stage training curriculum by always mixing data from other stages with a certain probability ($p = 0.3$). This updated training curriculum yields similar performance and enables effective control over the switch between latent and language reasoning.

**Metrics.** We apply two sets of evaluation metrics. One of them is based on the correctness of the *final answer*, regardless of the reasoning process. It is the metric used in the main experimental results above (Section 4.4). To enable fine-grained analysis, we define another metric on the *reasoning process*. Assuming we have a complete language reasoning chain which specifies a path in the graph, we can classify it into (1) **Correct Path**: The output is one of the shortest paths to the correct answer. (2) **Longer Path**: A valid path that correctly answers the question but is longer than the shortest path. (3) **Hallucination**: The path includes nonexistent edges or is disconnected. (4) **Wrong Target**: A valid path in the graph, but the destination node is not the one being asked. These four categories naturally apply to the output from COCONUT ($k = 0$) and *CoT*, which generate the full path. For COCONUT with $k > 0$ that outputs only partial paths in language (with the initial steps in continuous reasoning), we classify the reasoning as a Correct Path *if a valid explanation can complete it*. Also, we define Longer Path and Wrong Target for partial paths similarly. If no valid explanation completes the path, it's classified as hallucination. In *no-CoT* and COCONUT with larger $k$, the model may only outputs the final answer without any partial path, it falls into (5) **Correct Label** or (6) **Incorrect Label**. These six categories cover all cases without overlap.

## 5.2 INTERPOLATING BETWEEN LATENT AND LANGUAGE REASONING

Figure 5 shows a comparative analysis of different reasoning methods on ProsQA. As more reasoning is done with continuous thoughts (increasing $k$), both final answer accuracy (Figure 5, left) and the rate of correct reasoning processes ("Correct Label" and "Correct Path" in Figure 5, right) improve. Additionally, the rate of "Hallucination" and "Wrong Target" decrease, which typically occur when the model makes a wrong move earlier. This also indicates the better planning ability when more reasoning happens in the latent space.

A case study is shown in Figure 6, where *CoT* hallucinates an inexistent edge, COCONUT ($k = 1$) leads to a wrong target, but COCONUT ($k = 2$) successfully solves the problem. In this example, the model cannot accurately determine which edge to choose at the earlier step. However, as latent reasoning can avoid making a hard choice upfront, the model can progressively eliminate incorrect options in subsequent steps and achieves higher accuracy at the end of reasoning. We show more evidence and details of this reasoning process in Section 5.3 and 5.4.

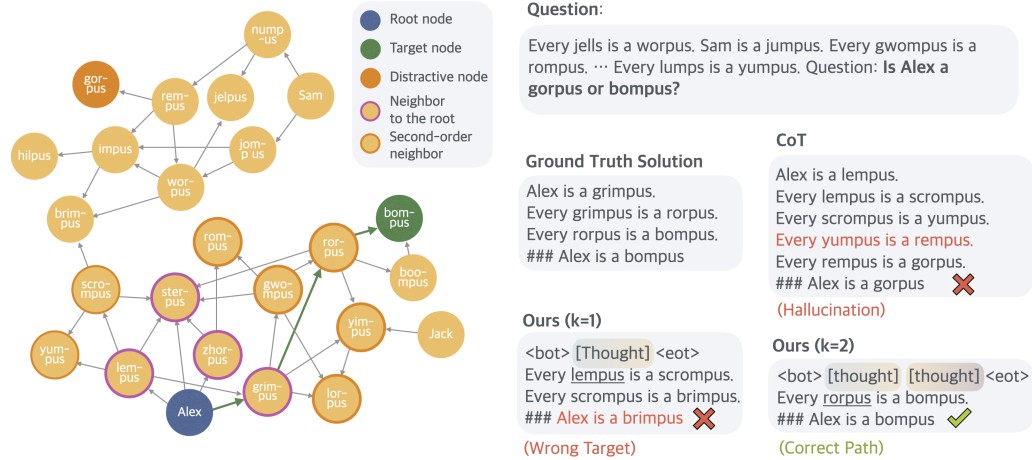

Figure 6: A case study on ProsQA. The model trained with *CoT* hallucinates an edge (*Every yumpus is a rempus*) after getting stuck in a dead end. COCONUT (k=1) outputs a path that ends with an irrelevant node. COCONUT (k=2) solves the problem correctly.

The comparison between *CoT* and COCONUT ($k = 0$) reveals another interesting fact: even when COCONUT is forced to generate a complete reasoning chain, the accuracy of the answers is still higher than *CoT*. The generated reasoning paths are also more accurate with less hallucination. From this, we can infer that the training method of mixing different stages improves the model's ability to plan ahead. The training objective of *CoT* always concentrates on the generation of the immediate next step, making the model "shortsighted". In later stages of COCONUT training, the first few steps are hidden, allowing the model to focus more on future steps. This is similar to the findings by Gloeckle et al. (2024), where they propose multi-token prediction as a new pretraining objective to improve the LLM's ability to plan ahead.

### 5.3 DISCOVERING THE LATENT SEARCH TREE

Given the intuition that continuous thoughts can encode multiple potential next steps, the latent reasoning can be interpreted as a search tree, rather than merely a reasoning "chain". Taking the case of Figure 6 as a concrete example, the first step could be selecting one of the children of *Alex*, i.e., *{lempus, sterpus, zhorpus, grimpus}*. We depict all possible branches in the left part of Figure 8. Similarly, in the second step, the frontier nodes will be the grandchildren of *Alex* (Figure 8, right).

Unlike a standard BFS that explores all frontier nodes uniformly, we show that the model learns to prioritize promising nodes while pruning others. We derive the model's preference by examining the its subsequent outputs in language. For instance, if we force the model to switch back to the language space after one latent thought ($k = 1$), it will predict the next step in the form of "every [Concept A] is a [Concept B]" as the next step. By measuring the probability of being filled in the position of [Concept A], we acquire a numeric value for each children of

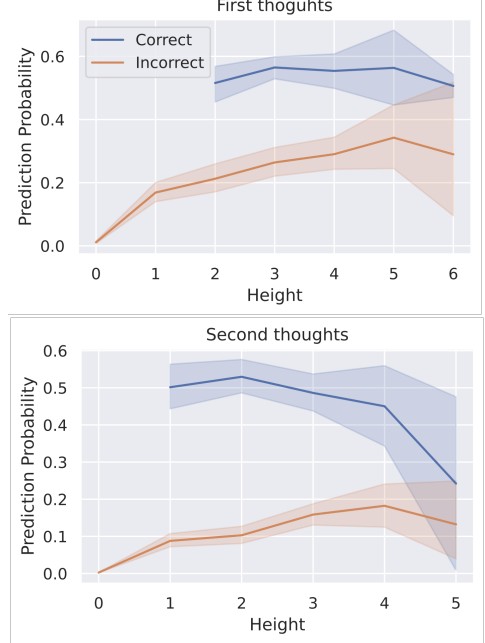

Figure 7: The correlation between prediction probability of concepts and their heights.

the root node *Alex* (Figure 8, left). Similarly, when we set $k = 2$, we can get the prediction probability of all the frontier nodes (The grandchildren of the root node *Alex*) in the second reasoning steps (Figure 8, right).

The probability distribution can be viewed as the model's implicit *value function*, estimating each node's potential to reach the target. As shown in the figure, "lempus", "zhorpus", "grimpus", and

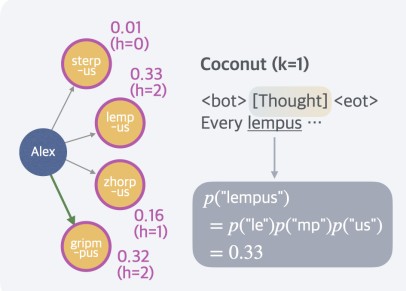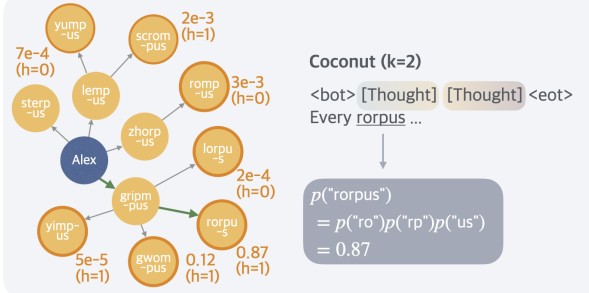

Figure 8: An illustration of the latent search trees. The example is the same test case as in Figure 6. The height of a node (denoted as $h$ in the figure) is defined as the longest distance to any leaf nodes in the graph. We show the probability of the first concept predicted by the model following latent thoughts (e.g., "lempus" in the left figure). It is calculated as the multiplication of the probability of all tokens within the concept conditioned on previous context (omitted in the figure for brevity). This metric can be interpreted as an implicit value function estimated by the model, assessing the potential of each node leading to the correct answer.

"sterpus" have a value of 0.33, 0.16, 0.32, and 0.01, respectively. This indicates that in the first continuous thought, the model has mostly ruled out "sterpus" as an option but remains uncertain about the correct choice among the other three. In the second thought, however, the model has mostly ruled out other options but focused on "rorpus".

### 5.4 WHY IS LATENT SPACE BETTER FOR PLANNING?

In this section, we aim to answer the question about why latent reasoning is better at planning, based on the search tree perspective and value function defined above. Referring to our previous example, a key distinction between *"sterpus"* and the other three options is that *"sterpus"* is a leaf node (Figure 6). This makes it immediately apparent as an incorrect choice, as it cannot reach the target node *"bompus"*. On the contrary, other nodes have more descendants to be explored, making them harder to evaluate. We measure each node's height (the shortest distance to any leaf nodes) as a proxy for its remaining exploratory potential. Based on this case, a natural hypothesis is that *the lower a node is, the easier it is to accurately estimate its value.* Indeed, here the model is mostly uncertain between "grimpus" and "lempus", both with a height of 2, which is higher than the other candidates.

To test this hypothesis, we analyze the correlation between the prediction probability and node height on the first and second latent steps across the test set. Figure 7 reveals a clear trend: the model effectively differentiates between correct and incorrect nodes when their heights are low, assigning a small value to incorrect nodes and a larger value to correct ones. However, as node heights increase, their gap narrows, indicating it's more challenging for the model to evaluate them accurately.

This empirical observation supports the idea that postponing definite decisions with latent thoughts is beneficial. As the latent search tree expands (through using more latent reasoning steps), the search frontier is pushed closer to the leaf nodes. Figure 7 confirms this, showing a larger gap between values of correct and incorrect nodes in the second step (lower figure) than in the first (upper figure). Therefore, more latent reasoning steps reduce the decision-making difficulty, allowing LLMs to make more accurate choices.

## 6 CONCLUSION

In this paper, we presented COCONUT, a novel paradigm for reasoning in continuous latent space, aimed to address the inherent inefficiencies associated with traditional language-based reasoning in large language models. Through extensive experimentation on various datasets, we demonstrated that COCONUT significantly enhances LLM reasoning capabilities. Notably, our detailed analysis highlighted how an unconstrained latent space allows the model to develop an effective reasoning pattern similar to BFS. We anticipate that our findings will inspire further research into latent reasoning methods, contributing to the development of more intelligent machine reasoning system.

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

| # Nodes | # Edges | Len of Shortest Path | # Shortest Paths |
|---------|---------|----------------------|------------------|
| 23.0    | 36.0    | 3.8                  | 1.6              |

Table 2: Statistics of the graph structure in ProsQA.

| Dataset  | Training | Validation | Test |
|----------|----------|------------|------|
| GSM8k    | 385,620  | 500        | 1319 |
| ProntoQA | 9,000    | 200        | 800  |
| ProsQA   | 17,886   | 300        | 500  |

Table 3: Statistics of the datasets.

# A  DATASETS

## A.1  CONSTRUCTION OF PROSQA

To construct the dataset, we need to define a set of entities, which are typical names like "Alex", "Jack", etc. We also define a set of concepts, which are fictional words like "lorpus", "rorpus", etc., following Saparov & He (2022).

The desired problem form is "Is [Entity] a [Concept A] or [Concept B]?". Assume the correct answer is [Concept A], we will need to construct a graph, so that we can find a path between [Entity] and [Concept A], and make sure [Entity] and [Concept B] are not connected.

The overall idea to build the DAG is to gradually add more nodes. Every time a new node comes in, we randomly add edges from existing nodes to the new node. We first sample the in-degree following a Poisson distribution with a mean equal to $1.5$, then sample the parents for this node. In this process, we need to make sure that any entity or concept cannot be the ancestor of both [Concept A] and [Concept B], in order to make a valid binary choice problem. Besides, we want to keep the family of [Concept A] and [Concept B] of similar sizes, otherwise the model may learn shortcuts.

Therefore, we implement a graph construction pipeline as follows: First, we initialize two nodes with labels 1 and 2. Then, for each new node, there is a probability $p$ ($p = 0.35$) that it can only accept edges from nodes with label 1; and another probability $p$ ($p = 0.35$) that it can only accept edges from nodes with label 2; otherwise the node can accepts edges from any nodes. After sampling the incoming edges for the node, it will be assigned a label: 1 if all the parent nodes have label 1; 2 if all the parent nodes have label 2; 3 if there are both parent nodes with label 1 and 2; 0 if there are no parent nodes or all parent nodes are labeled 0.

All nodes without parents will be assigned an entity name, while others are given a concept names. These form the known conditions. To get the question, we use the first node as the [Entity], a node labeled with 1 as [Concept A], a node labeled with 2 as [Concept B]. The construction will ensure there is always a path from [Entity] to [Concept A] but not [Concept B]. We will find the [Concept A] and [Concept B] that makes the reasoning chain relatively long. Note that after rendering the graph into natural language, we will permute the position of [Concept A] and [Concept B] randomly. Given the symmetry of label 1 and 2, there is no risk for shortcut in the position of choice.

The statistics of the resulting dataset is listed in Table 2.

## A.2  STATISTICS

We show the size of all datasets in Table 3.

# B  PARALLELISM OF LATENT TREE SEARCH

Figure 9 presents an analysis of the parallelism in the model's latent reasoning across the first and second thoughts. For the first thoughts (left panel), the cumulative values of the top-1, top-2, and top-3 candidate nodes are computed and plotted against their respective percentiles across the test set.

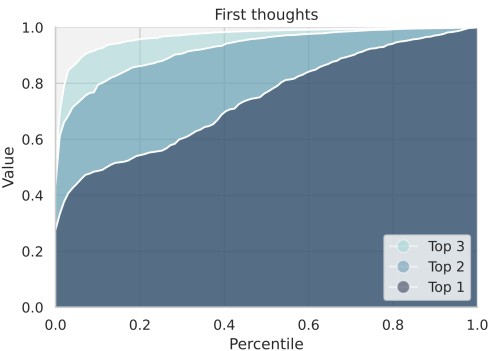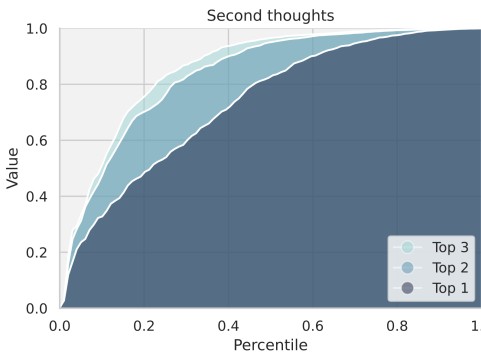

Figure 9: Analysis of parallelism in latent tree search. The left plot depicts the cumulative value of the top-1, top-2, and top-3 candidate nodes for the first thoughts, calculated across test cases and ranked by percentile. The significant gaps between the lines reflect the model's ability to explore alternative latent thoughts in parallel. The right plot shows the corresponding analysis for the second thoughts, where the gaps between lines are narrower, indicating reduced parallelism and increased certainty in reasoning as the search tree develops. This shift highlights the model's transition toward more focused exploration in later stages.

The noticeable gaps between the three lines indicate that the model maintains significant diversity in its reasoning paths at this stage, suggesting a broad exploration of alternative possibilities. In contrast, the second thoughts (right panel) show a narrowing of these gaps. This trend suggests that the model transitions from parallel exploration to more focused reasoning in the second latent reasoning step, likely as it gains more certainty about the most promising paths.

