# OpenReview forum: "Training Large Language Model to Reason in a Continuous Latent Space"
_ICLR.cc/2025/Conference — Submitted to ICLR 2025_

### Official Review · Reviewer_h5iq · 2024-11-02

**Soundness:** 3
**Presentation:** 3
**Contribution:** 2
**Rating:** 6
**Confidence:** 4

**Summary:**

The paper proposes a training scheme to teach language models to perform chain-of-thought reasoning using continuous intermediate steps (as opposed to tokens as in standard CoT). The training scheme works by iteratively distilling a prefix of the reasoning steps used in standard chain of thought into intermediate continuous 'tokens'. These intermediate continuous tokens are optimized end-to-end, so that a $k$-token long continuous reasoning step would require differentiation through $k$ iterative forward passes. They find that this technique is worse than CoT for GSM8K, comparable for ProntoQA, and outperforms CoT for a custom designed version of ProntoQA that is more challenging.

**Strengths:**

- Continuous latents are an interesting idea that may lead to more efficient reasoning
- The analysis done to understand the latent reasoning process is clever

**Weaknesses:**

- Method is only able to beat vanilla CoT on ProsQA, and is significantly worse than CoT on GSM8K
- Comparison with CoT conflates a few factors:
	- One factor is the training supervision: standard CoT is standard imitation learning, whereas the proposed method backpropagates through intermediate tokens.
	- Another factor is the use of latent vs discrete representations of intermediate reasoning
	- These two factors can be disentangled by either modifying proposed method to not backpropgate through various layers, or to somehow enable CoT to be backpropagated through
- Sec. 5 did not make it clear enough that the model being interpreted is not the same as the models used in the results for section. It is possible that the modified training procedure in sec. 5 (i.e. mixing in different stages) could lead to a qualitatively different model, both in terms of performance and reasoning mechanisms. I suggest highlighting this difference more clearly at the start of sec. 5
- The body of section 5.3 is rather poorly written. I am largely confused about what your hypotheses are, and experiments exactly you are conducting. This is a shame --- I think it likely contains interesting findings, but I cannot understand it.

**Questions:**

- Can you elaborate more on section 5.3? What is the overall claim, what are the experiments done to support that, etc.
- The interpretation that latent thoughts let you search in parallel is interesting. Do you have a sense of how parallel this can be, or how you can quantify this?

---

> ### Author Response · Authors · 2024-11-22
> **Response to Reviewer h5iq - Part I**
>
> We thank the reviewer for their thoughtful comments and appreciate the opportunity to address the concerns raised. Below, we respond to each point in detail and provide clarifications regarding our work.
>
> ## Scope of Work and Performance in Comparison with CoT
>
> Please kindly refer to our general response regarding the scope of this work and its comparison with CoT. Our primary objective is not to position COCONUT as a direct replacement for CoT but to explore the core attributes of latent space reasoning and analyze its potential as a novel paradigm. It’s also worth noting that COCONUT achieves a relative improvement of 13.7% over the latest implicit CoT reasoning method [1] on GSM8k, demonstrating progress in this emerging area.
>
> ## Conceptual Connection to CoT
>
> Thank you for your insightful comments. We agree that the differences between our method and CoT can be understood along two key dimensions:
>
> - **Learning Objective**: Imitation learning on the reasoning process vs. optimization towards future outcomes (e.g., the likelihood of future steps and the final answer)
> - **Reasoning Representation**: Discrete vs. continuous representations for intermediate reasoning steps.
>
> Breaking Down the Combinations:
> - **Imitation learning + discrete representation**:
>   This corresponds to CoT, where reasoning steps are explicitly represented as language tokens and supervised through imitation learning.
> - **No imitation learning + continuous representation**:
>   This is the approach introduced in our work, where reasoning is performed in a continuous latent space, and the objective is to optimize the likelihood of future steps and the final answer.
> - **Imitation learning + continuous representation**:
>   This approach is not feasible due to the absence of ground truth labels for continuous reasoning representations. While previous work [1] attempted to train smaller models to imitate the hidden states of larger models for reasoning via knowledge distillation, their performance remains limited. In our study, we have compared COCONUT against its follow-up work [2] as a stronger baseline.
> - **No imitation learning + discrete representation**:
>   This combination is particularly challenging due to the non-differentiable nature of discrete tokens. Traditional methods aim to solve this with techniques such as Gumbel-softmax [3] to enable backpropagation, or reinforcement learning (RL) [4, 5] to optimize discrete objectives. However, these approaches have only become practical with the advent of powerful pre-trained models. Moreover, most existing RL methods require supervised fine-tuning (which is CoT training in our setting) before applying RL [6, 7]. At the initial stages of this project, we experimented with RL-based methods for reasoning using discrete tokens. However, we observed poor performance on GSM8k with GPT-2, often resulting in near-zero accuracy. This motivated our shift toward exploring continuous representations for reasoning, as they are inherently differentiable and offer a promising alternative for effective training.
>
> ## Different Model in Section 5
> We appreciate your concern and would like to clarify our intent. Our primary goal is to explore and analyze latent space reasoning versus language space reasoning, rather than advocating for any specific training schedule. As such, while the model used in Section 5 was not trained under exactly the same settings as the one in Section 4, we believe the results and analysis remain consistent with our objectives and claims.
>
> We would also like to highlight that the accuracy of the model used in Section 5 is comparable to that of the model evaluated in Section 4 (95.8% and 97.0% on ProsQA, as shown in Figure 5 and Table 1), while both are significantly higher than CoT (77.5%, as reported in Table 1). This consistency indicates that the reasoning patterns observed in Section 5 are likely robust and universal across these two different but closely related training methods.
> We acknowledge your feedback and will revise the paper to explicitly clarify this point at the start of Section 5, ensuring there is no ambiguity regarding the relationship between the models and analyses across different sections. Thank you for bringing this to our attention!

---

> > ### Author Response · Authors · 2024-11-22
> > **Response to Reviewer h5iq - Part II**
> >
> > ## Clarification on Section 5.3
> > We apologize for any confusion and have updated the section in the paper for better clarity (the new version has been uploaded) We have reorganized the original Section 5 into 4 subsections. Below is a brief summary:
> >
> > - Section 5.1 describes the experiment settings, including how we train the model and the definition of new metrics.
> > - In Section 5.2, we control the number of continuous thoughts to interpolate between fully latent reasoning and fully language reasoning. We find that when more reasoning is conducted in latent space, the final answer accuracy and the accuracy of reasoning paths are both higher.
> > - To understand these results, Section 5.3 introduces a method to represent the latent reasoning into a search tree, and defines the model’s implicit value function to evaluate each node.
> > - Based on the search tree perspective, Section 5.4 further explains why latent reasoning can be advantageous.
> >
> > The outline of the new section 5.4 (originally the later part of section 5.3) is shown below:
> > - **Hypothesis**: Our intuitive hypothesis is that making decisions about the next step is easier for LLMs when the nodes being considered are closer to leaf nodes in the graph. At these points, it becomes more apparent whether a given node will lead to the target node. In contrast, when nodes are farther from the leaf nodes, it is challenging to evaluate whether they lie on the correct reasoning path.
> >
> > - **Empirical Validation**: To validate this hypothesis, we analyzed LLMs’ ability to evaluate nodes based on their height (the shortest distance to any leaf node). We plotted the model’s evaluation (value function) relative to the height of the nodes, and the results clearly support our hypothesis:
> >   - At lower heights (closer to leaf nodes), LLMs can effectively distinguish between correct and incorrect nodes, assigning higher scores to correct nodes and lower scores to incorrect ones.
> >   - At greater heights (farther from leaf nodes), LLMs struggle to make distinctions between correct and incorrect nodes.
> >
> >   This empirical observation supports the idea that postponing definite decisions with latent thoughts is beneficial. As the latent search tree expands (through using more latent reasoning steps), the frontier nodes are pushed closer to the leaf nodes. This reduces the decision-making difficulty, allowing LLMs to make more accurate choices.
> >
> > We hope this clarifies the purpose and findings of Section 5.3 and strengthens the understanding of latent reasoning’s advantages. Meanwhile, we are still working on quantify the degree of parallelism in the tree search to make the analysis more comprehensive. We will update the response as soon as we get some new results.
> >
> > ---
> >
> > ## Reference
> >
> > [1] Deng et al., 2023, “Implicit Chain of Thought Reasoning via Knowledge Distillation”
> >
> > [2] Deng et al., 2024, “From Explicit CoT to Implicit CoT: Learning to Internalize CoT Step by Step”
> >
> > [3] Kusner et al., 2016, “GANS for Sequences of Discrete Elements with the Gumbel-softmax Distribution”
> >
> > [4] Li et al., 2016, “Deep Reinforcement Learning for Dialogue Generation”
> >
> > [5] Rennie et al., 2016, “Self-critical Sequence Training for Image Captioning”
> >
> > [6] Ouyang et al., 2022, “Training language models to follow instructions with human feedback”
> >
> > [7] Shao et al., 2024, “DeepSeekMath: Pushing the Limits of Mathematical Reasoning in Open Language Models”

---

> > > ### Comment · Reviewer_h5iq · 2024-11-23
> > >
> > > Thanks for the response. The updated section 5 is much more readable.
> > >
> > > Despite not having a clear advantage over vanilla CoT for the more realistic GSM8k, the authors still presents an interesting architecture and training objective. Most valuable to me is the analysis in Sec. 5 that shows that the model can search in parallel. As such, I will increase my score.

---

> > > > ### Author Response · Authors · 2024-11-28
> > > >
> > > > Thank you for the encouraging and positive feedback! We have added an analysis of the parallelism of latent tree search in Appendix B.
> > > >
> > > > Our current work primarily focuses on analyzing the core attributes of latent reasoning. Building on this foundation, future research will explore the LLM pre-training with latent thoughts, which we believe will make latent reasoning more generally useful on realistic datasets.
> > > >
> > > > Thanks again for your valuable feedback.

---

### Official Review · Reviewer_G21p · 2024-11-02

**Soundness:** 4
**Presentation:** 3
**Contribution:** 3
**Rating:** 6
**Confidence:** 4

**Summary:**

This paper introduces a new LLM training/inference paradigm called Coconut, which includes a reasoning step in the continuous space. By training the model to predict hidden representations instead of actual tokens, the proposed solution could "think" before giving the final outputs without generating the CoT tokens. This paper also conducted experiments on three datasets: GSM8K, ProntoQA, and ProsQA to demonstrate that the proposed method is a more effective CoT parading than baselines such as iCoT and Pause Token.

**Strengths:**

1. The paper is clearly written and easy to follow.
2. The proposed method is simple and yet effective a well-defined tasks.
3. The experiment is solid and comprehensive.

**Weaknesses:**

The proposed method's main weakness is that it requires training on specific datasets. This makes it a task-specific solution rather than a general-purpose LLM solution, which diverges from the current research trend. Specifically, through the proposed training, I agree that the model could learn to compress the "thinking steps" into hidden representations. However, such representations could not be generalized to the general domain, which is an advantage of the CoT method. For example, I suspect that simply changing the prompt of asking the questions might significantly influence the final performance.

**Questions:**

1. Why do you only evaluate the efficiency in terms of tokens? Clock-time might be a better metric since you use the same foundation model as the baselines.
2. The analysis of the impact of thoughts per step in Figure 3 is good. How about more thoughts on the GSM benchmark?
3. The performance conclusion on different benchmarks seems to diverge. Can you summarize what is the best task scenario of applying the proposed method?

---

> ### Author Response · Authors · 2024-11-22
> **Response to Reviewer G21p**
>
> We thank the reviewer for their detailed feedback and suggestions. Below, we address the raised concerns and clarify the contributions of our work.
>
> ## The scope of paper and general-purpose reasoning method
>
> As noted in our general response, the primary focus of this paper is to explore the core attributes of latent space reasoning and analyze its potential as a novel paradigm. While our method does require specific training datasets, this is a necessary step to investigate the fundamental feasibility of reasoning in a latent space. Importantly, the CoT method reported in our paper is also fine-tuned on specific training datasets. Though our approach may "diverge from the current research trend" of general-purpose LLM solutions, we view this as an opportunity to address a more fundamental problem, rather than building upon existing language space reasoning paradigms. This aligns with the long-term goal of enhancing machine reasoning capabilities.
>
> ## Efficiency Metric
> Our original motivation is to minimize external factors that could influence the comparison. For instance, existing libraries may have been specifically optimized for text generation with latest ML system techniques, whereas our implementation of continuous thought generation may not yet be fully optimized. Fundamentally, the computational cost of generating a continuous thought is inherently lower than that of generating a token. As such, we use the number of tokens as a conservative estimate of its efficiency.
>
> Below, we supplement the clock-time comparison. The reported numbers are the average time of inference on one test case (in second), with batch size = 1, on an A100 GPU. We use the standard `generate` method in the transformers library for no CoT and CoT.
>
> ||GSM8k|ProntoQA|ProsQA|
> |-|-|-|-|
> |no CoT|0.03|0.03|0.08|
> |CoT|0.26|0.85|0.47|
> |Ours|0.09|0.11|0.15|
>
> The clock time is generally proportional to the number of new generated tokens reported in Table 1. We will add the new results to our paper.
>
> ## Impact of Prompt Variations
>
> The GSM8k dataset includes diverse math problems **expressed in free text**, and therefore we believe the improved accuracy of our model can indicate its robustness to certain prompt variations. This experimental setting also closely follows previous works [1, 2]. Could you clarify the specific types of prompt changes you are interested in?
>
> ## More thoughts on GSM8k
>
> Please kindly refer to our supplemented experiments and discussions in the general response.
>
> ## Divergent Benchmark Results
>
> - Real-World vs. Synthetic Domains:
>   GSM8k represents a real-world, open-domain question-answering task. Unlike the synthetic datasets used in our study, it demands more complex contextual understanding and modeling, which can place greater demands on computational capabilities. This hypothesis is supported by the observation that Coconut outperforms all other latent reasoning methods, and its accuracy steadily improves as the number of thoughts per step ($c$) increases from 0 to 2. Additionally, GSM8k requires diverse commonsense and world knowledge. This may give CoT an advantage, as it aligns closely with the pretraining objectives of the underlying language model, enabling it to better leverage its knowledge compared to Coconut.
> - Planning Requirements:
>   Complex reasoning tasks often require the model to "look ahead" to determine whether a particular step is optimal (also known as planning). Among the datasets in our experiments, GSM8k involves grade-school-level math word problems that allow for intuitive judgment of the next reasoning step. Similarly, ProntoQA includes distracting branches of limited size, making it relatively straightforward to identify the correct next step. In contrast, ProsQA, based on a randomly generated Directed Acyclic Graph (DAG) structure, presents a significant challenge to the model's planning abilities. Our experimental results suggest that tasks requiring extensive planning benefit more from latent space reasoning (including Coconut, some of its variants, and iCoT) than from reasoning using language tokens (CoT). Further analysis of this phenomenon is provided in Section 5.
>
> These discussions can be found in lines 328–332 and 334–343 of the paper.
>
>
> We hope this response clarifies the scope and contributions of our work, and we appreciate the valuable insights provided by the reviewers. We plan to add all the new results and discussions to the paper.
>
> ## Reference
>
> [1] Deng et al., 2023, “Implicit Chain of Thought Reasoning via Knowledge Distillation”
>
> [2] Deng et al., 2024, “From Explicit CoT to Implicit CoT: Learning to Internalize CoT Step by Step”

---

> > ### Comment · Reviewer_G21p · 2024-11-28
> >
> > Thanks for your additional effort on the analysis. That makes the paper more complete. I have increased my overall rating to 6.

---

> ### Author Response · Authors · 2024-11-26
> **Looking forward to discussion**
>
> Dear Reviewer,
>
> We appreciate the time you took to review our paper and provide valuable feedback. We have carefully addressed your concerns in our rebuttal and would be grateful if you could take a moment to read it.
>
> In particular, we would like to draw your attention to Section 5 (updated for better clarity), which reveals an interesting tree search pattern in latent space. It's worth noting that other reviewers have highlighted this section as one of the most valuable contributions of our paper. We would greatly appreciate hearing your thoughts on this section and any feedback you may have.
>
> Thank you for your time and consideration.

---

### Official Review · Reviewer_qvXF · 2024-11-03

**Soundness:** 2
**Presentation:** 3
**Contribution:** 2
**Rating:** 5
**Confidence:** 4

**Summary:**

The paper proposes a new reasoning paradigm for language models, called COCONUT (Chain of Continuous Thought), which allows reasoning in a continuous latent space rather than traditional language-based chains of thought (CoT). By using the last hidden state as an input embedding for subsequent reasoning steps, COCONUT avoids language constraints, potentially enhancing reasoning efficiency.  The authors use a multi-stage curriculum training scheme that gradually shifts from language-based reasoning to continuous thoughts, helping models learn effective latent reasoning representations. Experiment results with pre-trained GPT2 on synthetical datasets show that COCONUT is most effective in planning intensive tasks.

**Strengths:**

1. The paper proposed a novel approach to study the potential of making all verbal CoT steps latent.

2. The experiment results show the potential of COCONUT to be useful in terms of improved generation efficiency.

**Weaknesses:**

1. The proposed method requires training with verbal CoT steps but the performance is significantly lower on GSM than vanilla CoT training. While the generated tokens are significantly fewer, it is unclear if the efficiency benefit overweight the performance loss.

2. A generation efficiency analysis is needed to show the potential efficiency benefit of the proposed method.

3. More number of thoughts per step should be shown in Figure 3 to get a clearer picture of the effect of the thought length.

3. The experiment setting is in general very synthetic, with all three datasets synthetically generated. Some real-world reasoning datasets are needed to show the practical effectiveness of the proposed method.

4. Only one small LLM is used in the experiment, GPT2. More recent LLMs should be used to validate the effectiveness of the method.

5. The latent tree search analysis concludes that latent CoT can do BSF does not make sense to me. It seems that you can do a similar analysis to verbal CoT too.

6. The proposed method directly feeds the last layer output representation as the input embeddings might introduce extra difficulties in learning as the mismatch in input and output latent space, even the input and output embedding matrices are tied in GPT2. I'd suggest adding a trainable linear projection before feed the output representation back to the LLM.

**Questions:**

See weaknesses.

---

> ### Author Response · Authors · 2024-11-22
> **Response to Reviewer qvXF**
>
> We thank the reviewer for their detailed feedback and suggestions. Below, we address the raised concerns and clarify the contributions of our work.
>
> ## The scope of this paper and comparison to CoT
> While the performance of Coconut is worse than CoT on GSM8k, we’d like to emphasize that it outperforms the latest implicit CoT reasoning method [1] by a relative improvement of 13.7%. We’d like to emphasize that the scope of this paper is to focus on analyzing the core attributes of latent space reasoning (please refer to our general response), rather than proposing an immediate replacement of CoT.
>
> ## Generation efficiency
> To address concerns about the efficiency-performance trade-off, we report the number of generated tokens as a conservative metric for generation efficiency (Table 1), given that the computation of a continuous thought is strictly less than that of a token. Additionally, we have supplemented this with clock-time comparisons. Please kindly refer to our response to G21p.
>
> ## More continuous thoughts in Figure 3; Results on larger models
> Please kindly refer to our supplemented experiments and discussions in the general response.
>
> ## Latent tree search analysis
> Latent reasoning offers unique advantages by enabling the model to encode multiple potential next steps simultaneously in its continuous thought representations. This contrasts with verbal CoT, which requires deterministic decisions at every step.
> We illustrate this distinction with a concrete example in Figure 6:
>
> - In verbal CoT, the model prematurely selects “lempus” as the next step, which is incorrect. In contrast, Coconut maintains a latent distribution representing multiple options, assigning similar scores to “lempus” (0.33) and “grimpus” (0.32).
> - By the second latent reasoning step, Coconut progressively eliminates incorrect options, assigning a high score (0.87) to “rorpus,” which is on the correct reasoning path.
>
> Therefore, we respectfully disagree with the statement that “a similar analysis can be done to verbal CoT”. While one could also construct a search tree for verbal CoT, the decoding process is essentially a preliminary **greedy search**, which fundamentally differs from latent reasoning's ability to perform **BFS**.
>
> We have reorganized Section 5 for improved clarity and logical flow to address this feedback. The new version has been uploaded.
>
> ## Trainable linear layer
> We appreciate the reviewer’s suggestion to include a trainable linear projection layer to address potential input-output mismatches. We conducted experiments with this design during the initial stages of our work, but the results showed minimal differences. Our hypothesis is that since we train all parameters of the LLM, other parameters adapt effectively to produce and process continuous thoughts, mitigating the mismatch.
>
> We hope this response clarifies the scope and contributions of our work, and we appreciate the valuable insights provided by the reviewers. We plan to add all the new results and discussions to the paper.
>
> [1] Deng et al., 2023, “From Explicit CoT to Implicit CoT: Learning to Internalize CoT Step by Step”

---

> ### Author Response · Authors · 2024-11-26
> **Looking forward to discussion**
>
> Dear Reviewer,
>
> We appreciate the time you took to review our paper and provide valuable feedback. We have carefully addressed your concerns in our rebuttal and would be grateful if you could take a moment to read it.
>
> In particular, we would like to draw your attention to Section 5 (updated for better clarity), which reveals an interesting tree search pattern in latent space. It's worth noting that other reviewers have highlighted this section as one of the most valuable contributions of our paper. We would greatly appreciate hearing your thoughts on this section and any feedback you may have.
>
> Thank you for your time and consideration.

---

> > ### Comment · Reviewer_qvXF · 2024-12-02
> > **The position of this paper is unclear**
> >
> > I increased my score to 5. I agree the tree search finding is interesting, but I do not think this grants the acceptance of the current paper.
> >
> > At the first read of the submission, I naturally assumed that this was a method paper that proposed a new training technique of LLMs that makes all CoT steps latent. From this perspective, the paper clearly blows the acceptance bar as the performance of the proposed method only significantly outperforms the CoT baseline on one synthetic dataset proposed by the authors (ProsQA). The proposed method is even significantly worse than CoT on GSM8K. The increased efficiency cannot make up for this huge drop.
> >
> > In the rebuttal, the authors argue that the paper is an analysis paper that shows interesting attributes of the latent CoT. In this case, the newly added analysis is not enough for an analysis paper: the tree-search analysis only uses one dataset (ProsQA), and mainly focuses on the first two reasoning steps. The whole tree search is only analyzed with one specific example.
> >
> > In summary, the better planning performance that might be achieved by the proposed latent CoT is only hypothetical, with one synthetic dataset. If the performance-wise advantage of latent CoT can only be proved as the field progresses, I would suggest the authors just revise and posit this paper as an analysis paper or even a theory paper, and show the value of latent CoT more concretely from the tree-search perspective.

---

> ### Author Response · Authors · 2024-12-04
> **Follow-up Response to Reviewer qvXF (Part I)**
>
> Thanks for acknowledging the tree search analysis is interesting and raising the score. We respectfully disagree with the dichotomy between method and analysis papers. By proposing a new training method, we believe this paper has provided **both strong empirical results and an insightful analysis** on latent space reasoning.
>
> ## Empirical Results
>
> We contend that the value of a method should not be assessed solely on whether it outperforms existing methods across all datasets. We believe our empirical results presented in this work are significant for several reasons:
>
> - Reasoning without generating a language reasoning chain is **an extremely challenging task**, especially on GSM8k. As a reference:
>   - The pause-token paper [1] does not compare to CoT throughout the paper. The proposed method achieves 8.5% accuracy on GSM8k, slightly improving over the no-CoT baseline, which yields 7.5% accuracy.
>   - The filler token paper [2] focuses on synthetic datasets, and the performance is still worse than CoT.
>   - The iCoT papers [3, 4] propose methods that match CoT performance on certain synthetic datasets but remain behind CoT on GSM8k. Furthermore, [4] highlights that even GPT-4, when prompted to reason without generating explicit language reasoning chains, achieves only a 44% accuracy on GSM8k, underscoring the difficulty of this task.
>
> - To the best of our knowledge, our paper is the first to demonstrate that latent reasoning can outperform explicit CoT on any individual dataset. On ProsQA—a dataset requiring strong planning abilities—COCONUT exceeds CoT by 25.2%. This provides **the first evidence of latent reasoning's advantages over CoT beyond efficiency**. Moreover, our analysis explores the underlying mechanisms driving this enhanced planning capability, offering a clear explanation of why latent reasoning excels in these scenarios.
>
> - On the challenging GSM8k dataset, our method clearly outperforms no-CoT baselines on GSM8K, and **achieves a 13% relative improvement over the latest baseline iCoT** [4]. Even though our performance is below that of models trained with explicit CoT, achieving 34% accuracy on GSM8k using a GPT-2 small model without generating explicit language reasoning chains is a non-trivial accomplishment. We also discussed the differences between datasets, explaining why latent reasoning faces greater challenges in outperforming language reasoning on GSM8k (Line 323).
>
> - In addition to overall performance, our empirical results reveal several interesting findings:
>   - **Enhanced Performance with Increased Latent Computing**: As demonstrated in Figure 3, employing more latent computation steps improves reasoning performance.
>   - **Interpretability of Latent Representations**: The latent representations are easily interpretable, and they are shown to encode multiple potential intermediate variables simultaneously (Figure 4).
>
> ## Analysis
>
> We believe our analysis is thorough and well-supported. To provide further clarification:
>
> - **Interpolating between latent and language reasoning (Section 5.2)**: By adjusting the number of continuous thoughts, we explore a spectrum from fully latent reasoning to fully language-based reasoning. The results show that increasing latent reasoning improves both final answer accuracy and reasoning path accuracy (Figure 5). This indicates that sometimes while the model initially struggles with decisions in early steps, it’s often able to make a correct decision later when earlier steps are conducted in latent space. Figure 6 provides an example to illustrate this behavior.
> - **Tree Search Interpretation (Section 5.3)**: To better understand this process, we interpret latent reasoning as a tree search within the problem graph of ProsQA, and also define the model’s implicit value function to evaluate each node.
> - **Advantages of Latent Reasoning (Section 5.4)**: Based on the search tree perspective, we further explain why latent reasoning can be advantageous. When a node is closer to the leaf nodes (indicating smaller room for exploration), the LLM can accurately estimate its value (Figure 7). As the latent search tree expands (through using more latent reasoning steps), the search frontier is pushed closer to the leaf nodes, thus making the decision easier for LLMs.

---

> ### Author Response · Authors · 2024-12-04
> **Follow-up Response to Reviewer qvXF (Part II)**
>
> To address the reviewer’s concerns:
>
> > The tree-search analysis only uses one dataset (ProsQA)
>
> There is a trade-off between the depth and the generality of an analysis. We used ProsQA for this analysis because its graph structure allows us to clearly define a search tree and apply metrics like “height” to quantify the exploration potential of nodes. These are not applicable to open-domain datasets like GSM8k. **This dataset enables us to conduct a focused, in-depth analysis of core planning abilities** while minimizing confounding factors such as language understanding.
>
> It’s worth noting that many “analysis papers” on LLM reasoning also focus on one synthetic datasets [5, 6, 7], like simple two-hop questions, but they still provide valuable insights into LLM reasoning by deeply exploring specific aspects of model behavior.
>
> > … and mainly focuses on the first two reasoning steps.
>
> We focus on the first two reasoning steps because latent reasoning tends to be most effective during the early stages. This is supported by two key observations:
>
> - The most significant improvement occurs when transitioning from Coconut (k=1) to Coconut (k=3) (Figure 5).
> - As discussed in Section 5.4, **nodes in the early steps are more challenging to evaluate, making parallel search particularly beneficial at this stage.**
>
> Focusing on these initial steps does not contradict our claims. On the contrary, it’s intended to enhance the clarity of our tree-search analysis and highlights the core strengths of latent reasoning.
>
> > The whole tree search is only analyzed with one specific example.
>
> **This is factually incorrect**. The majority of our analysis is conducted on the full test set, as shown in Figures 5, 7, 9. Figures 6, 8 present an example, which are for illustrative purposes and to better explain some key concepts, such as value function and height, as a preparation for the quantitative analysis on the entire test set.
>
> We also want to highlight that the examples in Figure 6 and 8 are not exceptional cases. They are just typical scenarios where increased latent reasoning enables problem-solving, while fewer latent steps do not. These examples do not compromise the generality of our findings, as they align with the broader statistical trends observed across the entire test set, e.g., the improvements from Coconut (k=1) to Coconut (k=3) shown in Figure 5.
>
> > In summary, the better planning performance that might be achieved by the proposed latent CoT is only hypothetical, with one synthetic dataset.
>
> We respectfully disagree with the characterization of the improved planning performance as "hypothetical." In addition to better overall accuracy on ProsQA, we have included extensive analysis (the whole section 5) to substantiate our claims, as summarized above.
>
> We would also like to emphasize that the ProsQA dataset is particularly well-suited for evaluating planning ability, as it is specifically designed to enlarge distractive branches, requiring the model to plan ahead before making decisions. In contrast, ProntoQA features similar question formats but with fewer branches. As shown in Table 1, the gap between latent reasoning and CoT is significantly larger on ProsQA compared to ProntoQA, further highlighting its planning advantages.
>
> ## Reference
>
> [1] Goyal et al. "Think before you speak: Training language models with pause tokens.", ICLR 2024.
>
> [2] Pfau et al., “Let's Think Dot by Dot: Hidden Computation in Transformer Language Models”, COLM 2024.
>
> [3] Deng et al., “Implicit Chain of Thought Reasoning via Knowledge Distillation”, arXiv preprint arXiv:2311.01460 (2023).
>
> [4] Deng et al., “From Explicit CoT to Implicit CoT: Learning to Internalize CoT Step by Step” arXiv preprint arXiv:2405.14838 (2024).
>
> [5] Yang, et al. "Do Large Language Models Latently Perform Multi-Hop Reasoning?." ACL 2024
>
> [6] Biran, et al. "Hopping Too Late: Exploring the Limitations of Large Language Models on Multi-Hop Queries." EMNLP 2024
>
> [7] Wang et al., “Grokked Transformers are Implicit Reasoners: A Mechanistic Journey to the Edge of Generalization” NeurIPS 2024

---

### Official Review · Reviewer_4Kef · 2024-11-04

**Soundness:** 2
**Presentation:** 3
**Contribution:** 2
**Rating:** 6
**Confidence:** 4

**Summary:**

This paper introduces COCONUT (Chain of Continuous Thought), a new reasoning framework for large language models (LLMs). The authors argue that traditional Chain-of-Thought (CoT) reasoning is inefficient and aims at coherence rather than reasoning. Thus, the motivation is to allow reasoning in a latent and continuous space. The method involves feeding the model's last hidden state directly back as the next input embedding, bypassing the need to generate intermediate reasoning steps into language tokens. COCONUT requires staged training to progressively introduce continuous thoughts in place of explicit language reasoning steps. Evaluations on logical reasoning show that COCONUT allows for greater efficiency by reducing token generation while improving accuracy. On the math reasoning task, COCONUT's performance is close to explicit CoT and outperforms prior methods on latent CoT. Through case studies, the paper shows how COCONUT represents multiple potential reasoning paths and progressively narrows down choices, avoiding premature commitments.

**Strengths:**

1. The proposed latent CoT framework is a novel contribution to reducing computing for more efficient CoT reasoning. The framework is well-motivated and theoretically sound.
2. The proposed framework outperforms prior implicit CoT methods. On logical reasoning tasks, it improves over explicit CoT with more efficient inference.
3. An interesting analysis of interpretability shows that latent CoT allows the model to progressively eliminate incorrect options in subsequent steps to achieve higher accuracy. This can potentially contribute to the ongoing research about scaling test-time compute.

**Weaknesses:**

1. The proposed latent CoT framework requires multi-stage training to learn the internalization process from explicit to latent, which requires a large number of epochs over the full model parameters. This training effort hinders the method's general usability, especially with longer reasoning chains and larger parameter sizes. It would be more informative to see a trade-off analysis with larger models (ideally above or within the 70B scale) to compare the required compute and efficiency for COCONUT (internalization training + inference) and explicit CoT (inference).
2. CoT is a pretty general-purpose inference strategy, so I don't think evaluating GSM8K and the two logical reasoning benchmarks is sufficient. The paper should at least include some tasks outside of the training distribution to show that COCONUT matches or is close to the generalizability of traditional CoT, which works out-of-box on many tasks for powerful foundation models. For example, common benchmarks like MMLU (regular & pro), TheoremQA, Human-Eval, ARC, and MedQA. Or hard challenges like "Game of 24", mini crosswords, and ChessBench.
3. GPT-2 does not seem sufficient for experiments involving a general tool like CoT. I would like to see more results with larger and more advanced models like the LLaMA and Qwen families.

**Questions:**

1. It looks like most of the CoT reasoning chains are under 100 tokens. I wonder if the internalization training will still work if the CoT chains are very long (some medical and pharmaceutical problems can have around 512-1024 tokens of the reasoning process). Will increasing the number of training stages handle long-context CoT chains?

---

> ### Author Response · Authors · 2024-11-22
> **Response to Reviewer 4Kef**
>
> Thank you for the thoughtful review and valuable feedback. Below, we address the key concerns raised and clarify our contributions.
>
> ## Scope of Work and comparison to CoT
>
> Please kindly refer to our general response. Our focus is on exploring latent space reasoning, especially its interesting attributes compared to language space reasoning, rather than framing this work as "Coconut vs. CoT." In case it wasn't immediately apparent, we would like to clarify that in this work, the CoT method refers specifically to **training** LLMs to generate natural language reasoning chains, rather than CoT **prompting** as proposed by Wei et al. [1].
>
> ## The Number of Training Epochs and Usability
>
> While the statement that our method “requires a large number of epochs” is true, it needs to be understood in the following contexts:
>
> - GPT-2 is a relatively small language model and needs more training to converge. As a reference, in our experiments on GSM8k, it requires about 10 epochs for CoT training to converge, while the baseline iCoT method [2] takes ~50 epochs. Coconut converges in 25–30 epochs in total.
>
> - Coconut’s learning objective does not align with existing pretraining paradigms, putting it at a disadvantage for continual training. However, as discussed in the general response, future work will explore pretraining paradigms optimized for latent space reasoning to mitigate this constraint and improve usability.
>
> ## Evaluation Domains and Generalization
>
> We acknowledge the importance of evaluating reasoning methods across diverse domains and out-of-distribution (OOD) tasks. However, we’d also like to point out that recent analysis [3] has shown that CoT can hardly help with any tasks other than math or symbolic reasoning. E.g. It cannot improve the performance on ARC, and on MMLU it only helps with math problems. [4] shows that CoT doesn’t help with Game-of-24 and Mini-crossword either.
>
> Given these observations, many recent works on LLM reasoning [5, 6] also focus on math reasoning exclusively, as it’s a representative form of reasoning requiring diverse skills and presents significant challenges. While extending evaluations to more tasks (including those requiring longer-context CoT chains) or under the OOD settings are important future directions, it is beyond the immediate scope of this paper, which focuses on analyzing the core attributes of latent space reasoning.
>
> We hope this response clarifies the scope and contributions of our work, and we appreciate the valuable insights provided by the reviewers. We will add all the new results and discussions to the paper.
>
> ## Reference
>
> [1] Chain-of-Thought Prompting Elicits Reasoning in Large Language Models
>
> [2] From Explicit CoT to Implicit CoT: Learning to Internalize CoT Step by Step
>
> [3] To CoT or Not to CoT? Chain-of-thought helps mainly on math and symbolic reasoning
>
> [4] Tree of Thoughts: Deliberate Problem Solving with Large Language Models
>
> [5] Teaching Large Language Models to Reason with Reinforcement Learning
>
> [6] Physics of Language Models: Part 2.1, Grade-School Math and the Hidden Reasoning Process

---

> > ### Comment · Reviewer_4Kef · 2024-11-29
> >
> > Thank you for your response! I was already positive about this paper. So, I will keep my current score.

---

### Author Response · Authors · 2024-11-22
**General Response - Part 1**

We are grateful to all reviewers for their detailed and constructive feedback. We are encouraged to see that reviewers find:

- Our proposed framework is a novel (4Kef, qvXF) and interesting (h5iq) method for latent space reasoning.
- The experiments are solid and comprehensive (G21p), where our method outperforms prior implicit CoT methods (4Kef), and shows the potential of better reasoning efficiency (4Kef, qvXF, h5iq)
- The interpretability analysis of latent reasoning is interesting (4Kef) and clever (h5iq).

We have addressed all the questions raised by reviewers with additional experiments or thorough clarifications via separate responses to each reviewer. There are some common concerns we would like to address in the general response.

## Scope of Work and Comparison to CoT
We appreciate the feedback regarding COCONUT in comparison to CoT. However, we want to clarify that our intent is not to position COCONUT as an immediate replacement for CoT. Rather, our goal is to explore the new direction of latent space reasoning (as emphasized in Introduction). Through this work, we want to highlight several novel findings on the promising attributes of latent reasoning, including the observation that chaining continuous thoughts enhances reasoning performance (Section 4) and the emergence of advanced reasoning patterns, such as tree search, within the latent reasoning framework (Section 5).

CoT, or language space reasoning, builds upon decades of research in the NLP and ML communities and has only recently gained mainstream traction thanks to highly capable pre-trained LLMs. In contrast, COCONUT explores a novel paradigm of reasoning in latent space, which, as a fundamentally different approach from language space reasoning, will require more time and research to mature into a practical and widely adopted technique. For example, there are no available model checkpoints pre-trained in a way similar to COCONUT at this point, which puts it at disadvantage in comparison to CoT. As a reference, previous research [1] also emphasizes that the “pause token” can only enhance the reasoning ability when pre-trained with randomly inserted “pause tokens”. A potential challenge is that this new field lacks established infrastructure for scaling compared to standard language modeling. Specifically, COCONUT requires recurrent generation of continuous thoughts during training, and thus is hard to take advantage of existing large-scale LLM training frameworks. Developing more efficient training infrastructures in the future is certainly possible (refer to Mamba [2] or other state-space models, which also involves recurrent computing during training). However, achieving this will require significant effort. We leave the ambitious goal of building a foundation model for latent reasoning as future work, aiming to achieve the level of generality and usability demonstrated by language space reasoning methods like CoT.

While it may take additional research efforts to develop a latent reasoning approach to match the maturity, generality, and performance of CoT, our work highlights its interesting attributes and the potential for future research. We are optimistic that as the field progresses, significant advancements in training methodologies, scaling techniques, and supporting infrastructure will unlock the full potential of latent reasoning.

## More continuous thoughts

We conducted additional experiments with c=3 under the same conditions as Figure 3. The results are as follows:

|c|acc. (%)|
|-|-|
|0|25.4 (+-0.09)|
|1|31.1 (+-1.44)|
|2|34.1 (+-1.51)|
|3|29.4 (+- 4.05)|

While increasing $c$ from 0 to 2 steadily improves accuracy, setting $c=3$ does not yield better accuracy and introduces higher variance across runs. Analysis of the training logs reveals that adding 3 continuous thoughts at once—particularly during the final stage switch—causes a sharp increase in training loss, leading to instability. The loss curve can be found in this [link](https://anonymous.4open.science/api/repo/coconut-rebuttal-CFD8/file/loss.png?v=36a0faa5).

Future work will explore finer-grained schedules, such as incrementally adding continuous thoughts one at a time while removing fewer language tokens (as in iCoT [3]). Additionally, combining language and latent reasoning—e.g., generating the reasoning skeleton in language and completing the reasoning process in latent space—could provide a promising direction for improving performance and stability.

---

> ### Author Response · Authors · 2024-11-22
> **General Response - Part 2**
>
> ## Scale up to large models
>
> We experimented with COCONUT using Llama 3.2-3B and Llama 3-8B with $c=1$, training for 3 epochs for Llama 3.2-3B and 2 epochs in Stage 0, followed by 1 epoch per subsequent stage. The results are as follows:
>
> | Model | no-CoT | Ours |
> |-|-|-|
> |Llama 3.2-3B|26.0|31.7|
> |Llama 3-8B|42.2|43.6|
>
> We still observe performance gains across both Llama 3.2-3B and Llama 3-8B models over the no-CoT baseline, though the gains are not as significant as those with GPT-2. A potential reason is that smaller models may particularly benefit from continuous thoughts to increase their effective network depth. Future work could explore improved training schedules, such as those proposed in iCoT, to enhance performance of larger models.
>
> We also want to highlight that our goal is not to replace CoT, but to explore the potential of latent reasoning. For instance, our results indicate that this approach might enable smaller language models to serve as a reasoning backbone, balancing performance and computational constraints by dynamically leveraging continuous thoughts. Moreover, the method's inherent advantages in planning and search (shown in Section 5) may inspire novel approaches to solving complex problems.
>
> ---
>
> We thank the reviewers once again for their thoughtful feedback and hope that the additional results and clarifications address their concerns.
>
> [1] Goyal et al., 2023, “Training Language Models With Pause Tokens”
>
> [2] Gu et al., 2023, “Linear-Time Sequence Modeling with Selective State Spaces”
>
> [3] Deng et al., 2023, “From Explicit CoT to Implicit CoT: Learning to Internalize CoT Step by Step”

---

### Public Comment · ~Jason_Rich_Darmawan1 · 2025-05-22
**Does setting `c > 1` enable the model to perform **beam search** in a single forward pass?**

Refer to Figure 7

[Beam Search](https://huggingface.co/blog/how-to-generate#beam-search)

---

### Meta-Review · Area_Chair_ik2d · 2024-12-17

**Metareview:**

This paper proposes fine-tuning a language model for chain-of-thought (CoT) style reasoning where the thought tokens are continuous instead of discrete. Experiments show that the proposed method underperforms baselines on GSM8K, matches baselines on ProntoQA and outperforms baselines on ProsQA which is a dataset introduced in this paper. The training procedure involves using progressively less thought supervision.

**Additional Comments On Reviewer Discussion:**

The reviewers found the paper to be clearly written and the problem to be important. One of the main concerns was the significant drop in performance in GSM8K. The authors assert that the paper should be viewed as introducing a new direction for latent thinking rather than as a replacement of CoT tuning. Reviewers appreciated the latent tree search analysis in section 5. Overall, I agree that the position of the paper needs to be more clearly stated in the paper itself which would require too significant of a change from the current version.

---

### Decision · Program_Chairs · 2025-01-22

Reject